# Effects of *mrpigG* on Development and Secondary Metabolism of *Monascus ruber* M7

**DOI:** 10.3390/jof6030156

**Published:** 2020-08-29

**Authors:** Li Li, Fusheng Chen

**Affiliations:** 1Hubei International Scientific and Technological Cooperation Base of Traditional Fermented Foods, Huazhong Agricultural University, Wuhan 430070, China; lili1991@webmail.hzau.edu.cn; 2College of Food Science and Technology, Huazhong Agricultural University, Wuhan 430070, China

**Keywords:** *Monascus ruber*, *mrpigG*, development, secondary metabolism

## Abstract

*Monascus* pigments (MPs) have been used as food colorants for several centuries in Asian countries and are now used throughout the world via Asian catering. The MP biosynthetic pathway has been well-illustrated, but the functions of a few genes, including *mrpigG*, in the MP gene cluster are still unclear. In the current study, in order to investigate the function of *mrpigG* in *M. ruber* M7, gene deletion (Δ*mrpigG*), complementation (Δ*mrpigG*::*mrpigG*) and overexpression (M7::*PtrpC*-*mrpigG*) mutants were successfully obtained. The morphologies and biomasses, as well as the MP and citrinin production, of these mutants were analyzed. The results revealed that the disruption, complementation and overexpression of *mrpigG* showed no apparent defects in morphology, biomass or citrinin production (except MP production) in Δ*mrpigG* compared with *M. ruber* M7. Although the MP profiles of Δ*mrpigG* and *M. ruber* M7 were almost the same—with both having four yellow pigments, two orange pigments (OPs) and two red pigments (RPs)—their yields were decreased in Δ*mrpigG* to a certain extent. Particularly, the content of rubropunctatin (an OP) and its derivative rubropunctamine (an RP) in Δ*mrpigG*, both of which have a five-carbon side chain, accounted for 57.7%, and 22.3% of those in *M. ruber* M7. On the other hand, monascorubrin (an OP) and its derivative monascorubramine (an RP), both of which have a seven-carbon side chain, were increased by 1.15 and 2.55 times, respectively, in Δ*mrpigG* compared with *M. ruber* M7. These results suggest that the MrPigG protein may preferentially catalyze the biosynthesis of MPs with a five-carbon side chain.

## 1. Introduction

*Monascus* spp., a type of medicinal and edible filamentous fungus, have been used to produce fermented foods and medicine in Asian countries, such as China, Japan and the Korean Peninsula, for nearly 2000 years [1,2]. At present, their fermented products, such as red fermented rice (RFR), also called red yeast rice, Anka, Hongqu, red Koji, and red mold rice, are widely used as food additives and nutraceutical supplements worldwide, owing to their production of beneficial secondary metabolites (SMs), mainly including *Monascus* pigments (MPs), monacolin K (MK) and γ-amino butyric acid [1,3,4], even though some strains of *Monascus* spp. may produce citrinin (CIT), a kind of nephrotoxic mycotoxin [5,6].

MPs are a complex mixture of compounds with a common azaphilone skeleton [7], and more than 110 MP components, most of which possess a five- or seven-carbon side chain, have been identified to date [8]. Despite their large-scale utilization and research, the biosynthesis of MPs remains inadequate to a certain extent. In the last 10 years, a unifying picture of the biosynthesis of MPs has emerged as a result of genome sequencing and the functional analysis of MPs’ biosynthetic gene clusters. In 2017, Chen et al. comprehensively proposed the biosynthetic pathway of MPs in *M. ruber* M7 [9]. Although the structures of MPs are extremely diverse, their biosynthesis employs a unitary trunk pathway that ushers intermediates towards the classical yellow (monascine, ankaflavin), orange (rubropunctatin, monascorubrin) and red (rubropunctamine, monascorubramine) pigments, and features a variety of shunt pathways branching off from the trunk pathway at highly reactive node compounds [9]. The functions of many genes in the MP gene clusters of *M. ruber* M7 have been investigated by gene modification. However, there are a few genes in the MP gene cluster, such as *mpigG*, *mpigH*, and *mpigI*, which have not been examined to date. In 2013, Balakrishnan et al. found that the *mppD* deletion in *M*. *purpureus* KACC (*mrpigG* in *M. ruber* M7) resulted in a significant decrease in the production of four MPs and speculated that MppD might be an obligate accessory protein involved in the product formation of a precursor of MPs, but no experimental data were provided [10]. In 2019, Chen et al. also guessed that MrPigG may contribute to the release of the highly reactive intermediate of MrPigA [8].

In this study, we first cloned the *mrpigG* gene from *M. ruber* M7, which encodes a protein sharing 93.01% amino acid sequence similarity with the amino oxidase/esterase of *M*. *pilosus* (Genbank: AGN71609.1) and 69.26% similarity with serine hydrolase FSH of *Penicillium occitanis* (Genbank: PCH03976.1). Subsequently, we constructed *mrpigG* deletion (Δ*mrpigG*), complementation (Δ*mrpigG*::*mrpigG*) and overexpression (M7::*PtrpC*-*mrpigG*) mutants of *M. ruber* M7, and analyzed their morphologies, biomasses, as well as their MP and CIT production. The results revealed that the *mrpigG* mutants have inconspicuous effects on their morphologies, growth and MP and citrinin production, except for MPs production in Δ*mrpigG*, which implies that MrPigG may prefer to biosynthesize MP compounds with a five-carbon side chain, compared to ones with a seven-carbon side chain. To our knowledge, this is the first report accounting for the mechanism used by genes in the MP gene cluster to control the length of the side chains of MPs.

## 2. Materials and Methods

### 2.1. Fungal Strains, Culture Media and Growth Conditions

*M. ruber* M7 (CCAM 070120, Culture Collection of State Key Laboratory of Agricultural Microbiology, Wuhan, China), which can produce MPs and CIT, but no MK, was used as a DNA donor and for transformation [11]. All strains used in this study are described in Table 1. Potato dextrose agar (PDA), malt extract agar (MA), Czapek yeast extract agar (CYA) and 25% glycerol nitrate agar (G25N) were utilized for morphological characterization [12,13]. PDA medium was used for the analyses of MPs and CIT production. G418 (Sigma-Aldrich, Shanghai, China) or hygromycin B (Sigma-Aldrich, Shanghai, China) was added to the media for transformant selection [14]. All strains were maintained on PDA slants at 28 °C.

### 2.2. Cloning and Analysis of the mrpigG Gene

A pair of primers, pigG F1–pigG R1 (Table 2), were designed to amplify the *mrpigG* gene, using Oligo 6 software (http://www.oligo.net/). PCR was carried out to amplify the *mrpigG* gene from the genome of *M. ruber* M7. Amino acid sequences encoded by *mrpigG* were predicted using Soft Berry’s FGENESH program (http://www.softberry.com), and the MrPigG functional regions were analyzed using the Pfam 33.1 program (http://pfam.xfam.org/). Homology of the deduced amino acid sequence was analyzed using the BLASTP program on the NCBI website (http://blast.ncbi.nlm.nih.gov/Blast.cgi).

### 2.3. Deletion, Complementation and Overexpression of the mrpigG Gene

To verify the function of *mrpigG*, the gene was deleted, complemented and overexpressed, according to the homologous recombination strategy, as previously described [14,15]. The *mrpigG* gene deletion cassette (5′UTR-*hph*-3′UTR), complementation cassette (5′UTR-*mrpigG*-*neo*-3′UTR) and overexpression cassette (5′UTR-*neo*-*PtrpC*-*mrpigG*-3′UTR) were constructed by double-joint PCR, and are shown schematically in Figures 1a, 2a and 3a, respectively. The relative primer pairs are shown in Table 2.

The gene disruption construct carried 5′ flanking regions (884 bp, amplified with the primer pair pigG d5F and pigG d5R), hygromycin B resistance gene (*hph*, 2137 bp, amplified from plasmid pSKH) with the primer pair hph F and hph R), and 3′ flanking regions (867 bp, amplified using the primer pair pigG d3F and pigG d3R), consecutively.

The gene complementation structure possessed 5′ flanking, together with *mrpigG* open reading frame (ORF) regions (1823 bp, amplified with the primer pair pigG c5F and pigG c5R), neomycin phosphotransferase resistance gene (*neo*, 1221 bp, amplified from plasmid pKN1 with the primer pair G418 F and G418 R) and 3′ flanking regions (860 bp, amplified with the primer pair pigG c3F and pigG c3R), consecutively.

The gene overexpression construct carried 5′ flanking regions (969 bp, amplified with the primer pair pigG ov5F and pigG ov5R), neomycin phosphotransferase resistance gene (*neo*, 1221 bp, amplified from plasmid pKN1 with the primer pair G418 F and G418 R), the *trpC* promoter (373 bp, amplified from plasmid pSKH using the primer pair PtrpC F and PtrpC R) and the *mrpigG* gene together with 3′ flanking regions (1621 bp, amplified with the primer pair pigG ov3F and pigG ov3R), consecutively.

Three abovementioned structures were separately cloned into pMD19-T (Takara, Dalian, China), then all the cloned fusion DNA fragments and plasmid pCAMBIA3300 were digested with *Kpn*I and *Hind*III, and were ligated by T4 DNA ligase to generate plasmids pCPIGG, pCCPIGG and pCOPIGG, respectively. Then three plasmids were separately transformed into *Agrobacterium tumefaciens* EHA105 using a freeze-thaw method. Then, three *A. tumefaciens* EHA105 clones containing pCPIGG, pCOPIGG were incubated for transformation with *M. ruber* M7 to yield the *mrpigG* disruption strains (Δ*mrpigG*), *mrpigG* and overexpression strains (M7::*PtrpC*-*mrpigG*), respectively. Meanwhile, the *A. tumefaciens* EHA105 clones containing pCCPIGG were incubated for transformation with the Δ*mrpigG* strain to generate the *mrpigG* complementation strains (*ΔmrpigG*::*mrpigG*).

### 2.4. Real-Time qPCR Analysis

The expression of *mrpigG* in the selected Δ*mrpigG, ΔmrpigG*::*mrpigG*, M7::*PtrpC*-*mrpigG* strain and the wild-type strain *(M. ruber* M7) was analyzed by real-time quantitative PCR analysis (RT-qPCR). One milliliter freshly harvested spores (10^5^ cfu/mL) of each strain were inoculated into PDA medium and incubated at 28 °C, and samples were taken every other day from the 3rd day to the 11th day. RT-qPCR was performed according to the method described by Liu et al. [14]. *GAPDH* was used as a reference gene. The primers used in these analyses are listed in Table 2.

### 2.5. MP and Citrinin Analyses

Previous studies have revealed that MPs accumulate mainly in the mycelia, where citrinin exists in media [16]. Thus, intracellular MPs and the extracellular citrinin were detected. One milliliter of freshly harvested spores (10^5^ cfu/mL) of each strain were inoculated on PDA plates covered with cellophane membranes, and incubated at 28 °C for 11 days. The samples were taken every 2 days from the 3rd day to the 11th day of culture to measure the MPs and CIT production [5]. Twenty milligrams of freeze-dried mycelia or media powder were suspended in 1 mL 80% (*v*/*v*) methanol solution, and subjected to 30 min ultrasonication treatment (KQ-250B, Kunshan, China).

Then, MPs were separated by an ACQUITY UPLC BEH C18 column (2.1 mm × 100 mm, 1.7 µm, Waters), and detected on a Waters ACQUITY UPLC I-class system by UV-Vis spectra (Waters, Milford, MA, USA). A gradient elution was performed with the mobile phase including solvent A (acetonitrile), solvent B (ultra-pure water) and solvent C (0.1% formic acid in water) with a flow rate of 0.3 mL/min and an injection volume of 2 µL. The gradient elution was performed as follows: firstly, Solvent A/B/C maintained 35:55:10 for 3 min, then the content was changed to 75:15:10 for 15 min, followed by adjusting to 90:0:10 for 5 min. Subsequently, the solvent was changed to 35:55:10 for 5 min. Finally, the column was equilibrated with 35:55:10 for 3 min.

The citrinin was detected on a Waters ACQUITY UPLC system by means of a fluorescence detector (Waters, Milford, MA, USA) A gradient elution was performed with the mobile phase including solvent A (0.1% formic acid in water) and solvent B (acetonitrile) with a flow rate of 0.3 mL/min and an injection volume of 2 µL. The gradient elution was performed as follows: Solvent A/B was first maintained at 90:10 for 3 min. Secondly, the content of solvent A was decreased from 90% to 30% for 7 min. Then, the amount of solvent A was decreased from 30% to 10% for 0.01 min and was maintained for 3 min. Finally, the amount of solvent A was increased from 10% to 90% for 3 min. The temperature of the chromatographic column and samples were maintained at 40 °C and 4 °C, respectively.

## 3. Results

### 3.1. Sequence Analysis of the mrpigG Gene in M. ruber M7

A 1.24-kb fragment containing the putative *mrpigG* homolog was successfully amplified from the genomic DNA of *M. ruber* M7. Sequence prediction of *mrpigG* by SoftBerry’s FGENESH program revealed that the putative *mrpigG* gene consists only of an 822 bp open reading frame (ORF) which consists of 1 exon and encodes 273 amino acids. A database search with the Pfam 33.1 program showed that MrPigG pertains to the serine hydrolase family (FSH). A database search with NCBI-BLAST demonstrated that the deduced 273-amino acid sequence encoded by *mrpigG* shares 93.01% similarity with the amino oxidase/esterase of *M*. *pilosus* (GenBank: AGN71609.1), 69.26% similarity with serine hydrolase FSH of *Penicillium occitanis* (GenBank: PCH03976.1), and 55.72% similarity with the serine hydrolase-domain-containing protein of *Pseudomassariella vexata* (GenBank: ORY68205.1).

### 3.2. Verification of the mrpigG Deletion, Complementation and Overexpression Strains

After the plasmid pCPIGG was transformed into *M. ruber* M7, transformants with hygromycin B resistance were obtained and verified by PCR, and three of 115 were transformant *mrpigG*-deleted strains. As shown in Figure 1c, no DNA band was amplified when genomic DNA of one *mrpigG* disruptant named Δ*mrpigG* was used as template with primers pigG F2-pigG R2, although a 0.5-kb product appeared using the genome of the wild-type strain *M. ruber* M7. A 2.14-kb fragment of the *hph* gene could be amplified from Δ*mrpigG* using primers hph F-hph R, while nothing was obtained from *M. ruber* M7. Meanwhile, amplicons of *M. ruber* M7 (2.28 kb) and Δ*mrpigG* (3.89 kb) differed in size when primers pigG d5F-pigG d3R annealing to homologous arms (5′ flanking and 3′ flanking regions) were used.

Three Δ*mrpigG*::*mrpigG* strains with G418 resistance were obtained. PCR analysis of one of the Δ*mrpigG*::*mrpigG* strains is shown in Figure 2c. A 0.5-kb product was amplified when the genomic DNA of Δ*mrpigG*::*mrpigG* was used as template with primers pigG F2-pigG R2, whereas no DNA band was amplified using the genome of Δ*mrpigG*. A 1.22-kb fragment of the *neo* gene could be amplified from Δ*mrpigG*::*mrpigG* using primers G418 F-G418 R, whereas nothing was obtained from Δ*mrpigG*. A 2.14-kb product of the *hph* gene could be amplified from Δ*mrpigG* using primers hphF-hphR, whereas nothing was obtained from Δ*mrpigG*::*mrpigG*. The *neo* fragment replaced the *hph* gene, which might reveal that it was a successful homologous recombination event. Meanwhile, amplicons of Δ*mrpigG*::*mrpigG* (3.9 kb) and Δ*mrpigG* (4.3 kb) differed in size when primers pigG d5F-pigG d3R, annealing to homologous arms, were used.

Seven M7::*PtrpC*-*mrpigG* strains with G418 resistance were obtained and checked by PCR analysis. As shown in Figure 3c, amplicons of *M. ruber* M7 and M7::*PtrpC*-*mrpigG* were equal in size when primers pigG F2-pigG R2 were used, and the size was 0.5 kb. A 1.22-kb fragment of the *neo* gene and a 0.37-kb product of the *PtrpC* gene could be amplified from M7::*PtrpC*-*mrpigG* using primers G418 F-G418 R and PtrpC F-PtrpR, whereas nothing was obtained from M7, respectively. Meanwhile, amplicons of M7::*PtrpC*-*mrpigG* (4.13 kb) and *M. ruber* M7 (2.56 kb) differed in size when primers pigG d5F-pigG d3R annealing to homologous arms were used, which proved that there was a single copy of the *PtrpC*-*mrpigG* overexpression construct integrated in M7::*PtrpC*-*mrpigG*.

### 3.3. RT-qPCR Analysis of ΔmrpigG, ΔmrpigG::mrpigG, M7::PtrpC-mrpigG and M. ruber M7

The transcription levels of the Δ*mrpigG* gene in the four strains—Δ*mrpigG*, Δ*mrpigG*::*mrpigG*, M7::Ptrp C-Δ*mrpigG* and the wild-type strain *M. ruber* M7—were analyzed by RT-qPCR. As shown in Figure 4, Δ*mrpigG* was deficient in the expression of the *mrpigG* gene, which further verified the success of the gene knockout. The tendency of *mrpigG* expression in Δ*mrpigG*::*mrpigG* was similar with that of *M. ruber* M7, and the average level of *mrpigG* expression in M7::*PtrpC-mrpigG* was higher than that of *M. ruber* M7, in spite of the 3rd day.

### 3.4. Morphologies and Biomasses of ΔmrpigG, ΔmrpigG::mrpigG, M7::PtrpC-mrpigG and M. ruber M7

Colonial and microscopic characteristics of *M. ruber* M7 were observed on the different media (PDA, CYA, MA, G25N) to investigate the influence of the *mrpigG* deletion, complementation and overexpression on developmental processes. The results showed that colonial morphologies (Figure 5a) and microscopic morphologies, including conidia and cleistothecia, of Δ*mrpigG*, Δ*mrpigG*::*mrpigG* and M7::*PtrpC*-*mrpigG* were not significantly different from those of *M. ruber* M7 on different culture plates (Figure 5b). Moreover, those four strains showed similar biomasses on PDA (Figure 5c).

### 3.5. MP and CIT Production Analysis of ΔmrpigG, ΔmrpigG::mrpigG, M7::PtrpC-mrpigG and M. ruber M7

Previous studies [5,9] have demonstrated that *M. ruber* M7 can produce MPs and CIT, but no MK, so the yields of the eight main MPs (four yellow pigments: monasfloure A, monascine, monasfloure B and ankaflavin; two orange pigments: rubropunctatin and monascorubrin; and two red pigments: rubropunctamine and monascorubramine) (Figure 6) and CIT in *M. ruber* M7 and its mutants were analyzed in this study to uncover the effect of *mrpigG* on SMs. As for the yellow pigment production, the levels of monasfloure A, monascine, monasfluore B and ankaflavin in Δ*mrpigG* were 30.7%, 40.1%, 67.2% and 94% of that in *M. ruber* M7, respectively (Figure 7a–d). For orange and red pigment production, the production of rubropunctatin (an orange pigment (OP)) and rubropunctamine (a red pigment (RP)) in Δ*mrpigG*, both of which contain a 5-carbon side chain, were 57.7% and 22.3% of those in *M. ruber* M7 (Figure 7e–f). On the other hand, monascorubrin (an OP) and monascorubramine (an RP) in Δ*mrpigG,* both of which have a 7-carbon side chain, were 1.15 and 2.55 times those of *M. ruber* M7 (Figure 7g–h).

As shown in Figure 7i, the tendency of all strains’ extracellular citrinin production was always increased from the 3rd day to the 11th day of cultivation in PDA media. During the early stage (3–7 days) the CIT production in *M. ruber* M7 was only 22.8–72.9% that of Δ*mrpigG*, whereas the other two mutants possessed similar yields to those of Δ*mrpigG*. At the end of the 11 days of fermentation, extracellular citrinin concentration of the four strains were similar to those of the wild-type strain *M. ruber* M7, at 1.88 ± 0.25 µg/mg citrinin.

## 4. Discussion

MPs with a common azaphilone skeleton are a large group of secondary metabolites produced via the polyketide pathway by filamentous fungi, mainly by *Monascus* spp.and *Penicillium* spp. [1,17]. MPs have been used in the food coloring, pharmaceutical, cosmetics, textile, printing and dyeing industries [3,18]. Recently, researchers proposed a MP biosynthetic pathway, in order to explore the generation of the bewildering diversity and complexity of MPs. To date, the functions of many genes in the MP gene clusters have been well investigated in *M. ruber* M7 and *M. purpureus* KACC42430, but there has been limited research on a few related genes, such as *mpigG*, *mpigH* and *mpigI* [8].

In current study, the generation of *mrpigG* mutants by homologous recombination, as shown in Figure 5 and Figure 7, demonstrated that *mrpigG* was a member of the MP gene cluster. Searched with the Pfam 33.1 program, the data showed that *mrpigG* belonged to the serine hydrolase family, which is a large, ubiquitous family of enzymes grouped based on their ability to perform hydrolysis reactions on a range of biological substrates, such as ester, thioester, amide and epoxide bonds in small molecules, peptides or proteins [19,20,21]. The majority of serine hydrolases (>60%) adopt an α,β-hydrolase fold and employ a Ser-His-Asp catalytic triad [21], but more than half of the serine hydrolases (>120 enzymes) remain poorly annotated, with no described physiological function or identified substrates [19,22].

In this study, the results showed that there was a 10–70% decrease in the four typical yellow pigments by *mrpigG* deletion. Simultaneously, the yields of typical orange pigments (OPs) and red pigments (RPs) revealed a completely different accumulation. In detail, compared with *M. ruber* M7, the yields of rubropunctatin (an OP) and its derivative rubropunctamine (an RP) produced by Δ*mrpigG* reduced to 57.7% and 22.3%, respectively. On the contrary, the production of monascorubrin (an OP) and its derivative monascorubramine (an RP) increase by 0.15-fold and 1.55-fold, respectively. These two OPs and RPs separately possessed the same azaphilone skeleton. In the same vein, rubropunctatin (OP) and rubropunctamine (RP) contained the −C_5_H_11_ side chain, wheres the monascorubrin (OP) and monascorubramine (RP) owned the −C_7_H_15_ side chain. The serine hydrolase encoded by *mpigG* specifically catalyzed the synthetic compounds with the −C_5_H_11_ side chain, and might show substrate specificity to a certain extent. Navia-Paldanius et al. delineated the substrate preferences of three serine hydrolases by analyzing accumulation of fatty acid esters in cell lysates individually overexpressing each of the enzymes in assay mixes containing different chain length and saturation acylglycerols [23]. Ritchie et al. elaborated that the human type II thioesterase TE2 (containing a Ser-Asp-His catalytic triad) preferred an engineered human acyl-ACP substrate releasing short chain fatty acids from full-length fatty acid synthase (FASN) during turnover. The ability of TE2 to release fatty acids from FASN was involved in polyketide and non-ribosomal peptide synthase synthases [24].

Taking OP and RP production into account, the function of *mrpigG* in the biosynthesis pathway of MPs was proposed, as shown in Figure 8. In the proposed pathway, the β-ketooctanoic or β-ketodecanoic acid moieties used for this acylation step (P2) are produced by a dedicated two-subunit fatty acid synthetase (MrPigJ and MrPigK in *M. ruber* M7) encoded in the MP gene clusters. The β-keto fatty acid, along with ACP, specially hydrolyzes by means of MrPigG, then the hydrolysates are transferred to the acyltransferase (MrPigD). The C-4 alcohol of the benzopyran in P1 is then acylated with a medium chain β-keto fatty acid by MrPigD to yield the putative intermediate P2, which has been characterized as a precursor of MPs [8,9,25]. MrPigG may act as a metabolic serine hydrolase, like type II TE proteins, promoting the release of the short chain compounds from the thioester moiety on ACP [10,26].

In conclusion, the *mrpigG* gene pertains to the MP biosynthetic gene cluster of *M. ruber* and plays an unignorable role in the biosynthesis of MPs. The disruption, complementation and overexpression of *mrpigG* had very little effect on the biomass, citrinin production and morphology of the strains. Compared with *M*. *ruber* M7, Δ*mrpigG* equally yielded eight typical pigments, but the production of four yellow pigments was partially restrained, and the synthesis of orange and red pigments depended on side chain specificity. The content of rubropunctatin (OP) and its derivative rubropunctamine (RP), both of which have a five-carbon side chain, were 57.7% and 22.3% of those in *M. ruber* M7. On the other hand, monascorubrin (OP) and its derivative monascorubramine (RP), both of which have a seven-carbon side chain, were 1.15 and 2.55 times of those found in *M. ruber* M7. This work will make some contribution to the study of the biosynthesis pathway of MPs in *M. ruber*.

## Figures and Tables

**Figure 1 jof-06-00156-f001:**
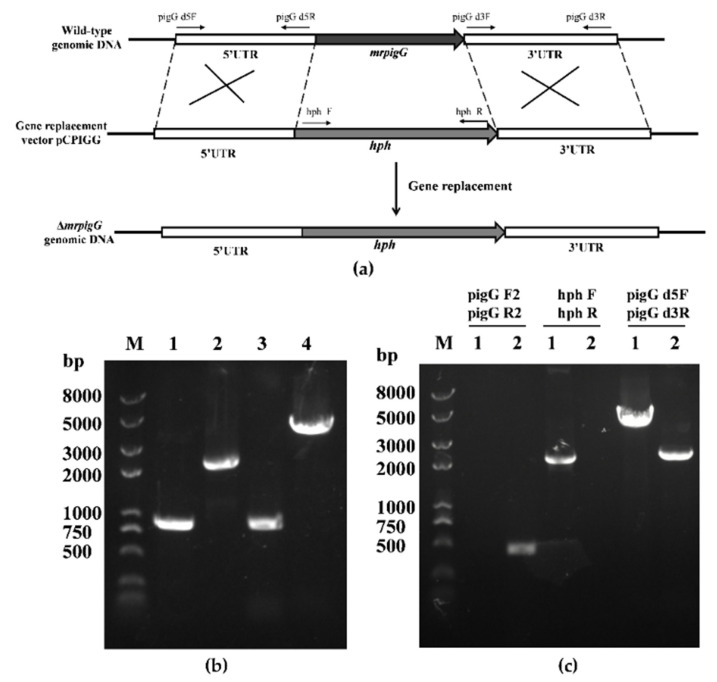
Deletion of *mrpigG* in *M. ruber* M7. (**a**) Schematic representation of the homologous recombination strategy yielding *mrpigG* deletion strains. (**b**) Construction of *mrpigG* disruption construct by double-joint PCR. Lane 1, 5′ flanking region of *mrpigG*; lane 2, hygromycin resistance cassette; lane 3, 3′ flanking region of *mrpigG*; lane 4, double-joint PCR product. (**c**) Confirmation of *mrpigG* homologous recombination events. Three primer pairs were used, and PCR amplifications showed distinct bands in different strains. Lane 1, the Δ*mrpigG* strain; lane 2, the wild-type strain.

**Figure 2 jof-06-00156-f002:**
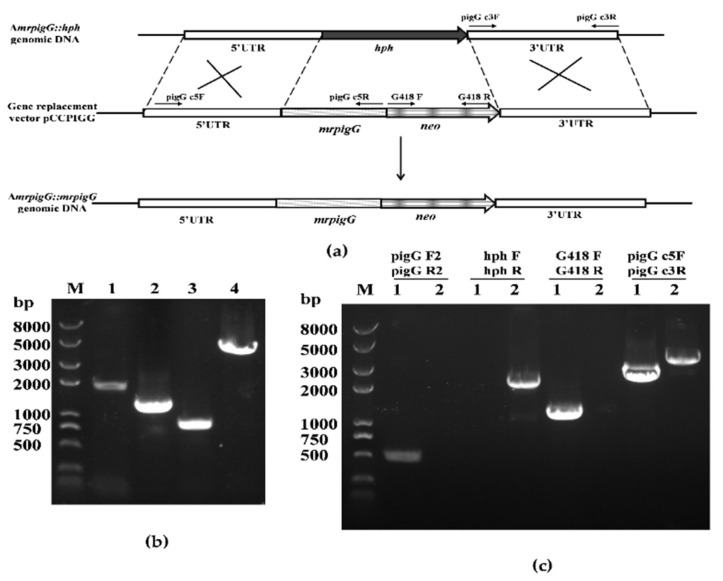
Complementation of *mrpigG* in *M. ruber* M7. (**a**) Schematic representation of the homologous recombination strategy to construct *mrpigG* complementation strains (**b**) Construction of *mrpigG* complementation cassette by double-joint PCR. Lane 1, 5′ flanking region of *mrpigG* plus *mrpigG* ORF regions; lane 2, G418 resistance cassette; lane 3, 3′ flanking region of *mrpigG*; lane 4, double-joint PCR product. (**c**) Confirmation of *mrpigG* homologous recombination events. Four primer pairs were used, and PCR amplifications showed distinct bands in different strains. Lane 1, the Δ*mrpigG*:: *mrpigG* strain; lane 2, the Δ*mrpigG* strain.

**Figure 3 jof-06-00156-f003:**
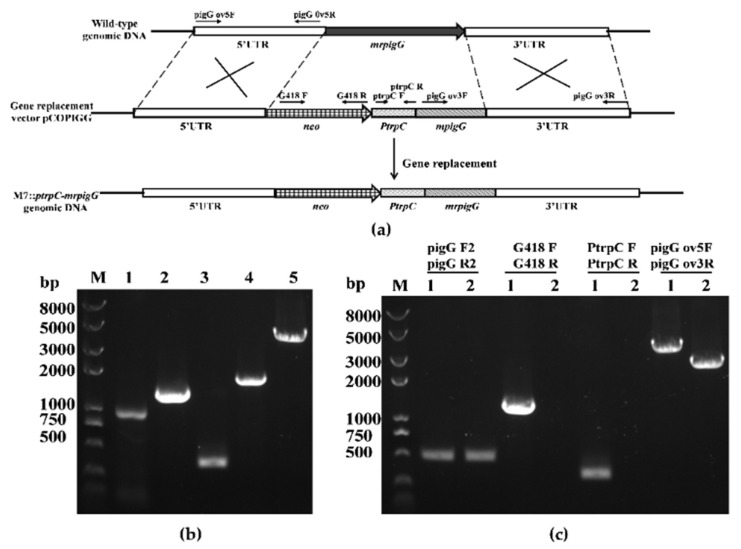
Overexpression of *mrpigG* in *M. ruber* M7. (**a**) Schematic representation of the homologous recombination strategy yielding *mrpigG* overexpression strains. (**b**) Construction of *mrpigG* overexpression construct by double-joint PCR. Lane 1, 5′ flanking region of *mrpigG*; lane 2, G418 resistance cassette; lane 3, the *trpC* promoter; lane 4, *mrpigG* ORF regions plus 3′ flanking region of *mrpigG*; lane 5, double-joint PCR product. (**c**) Confirmation of *mrpigG* homologous recombination events. Three primer pairs were used, and PCR amplifications showed distinct bands in different strains. Lane 1, the M7::*PtrpC-mrpigG* strain; lane 2, the wild-type strain.

**Figure 4 jof-06-00156-f004:**
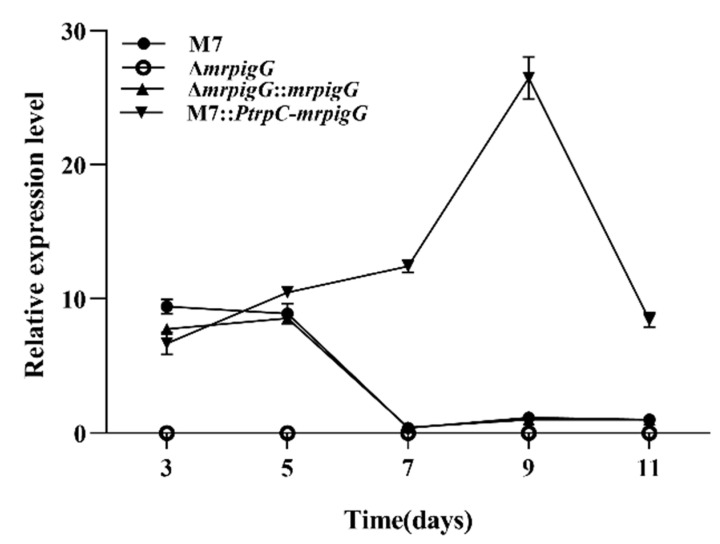
RT-qPCR analysis of the *mrpigG* gene in the *mrpigG* deletion strain, the *mrpigG* complementation strain, the *mrpigG* overexpression strain and the wild-type strain.

**Figure 5 jof-06-00156-f005:**
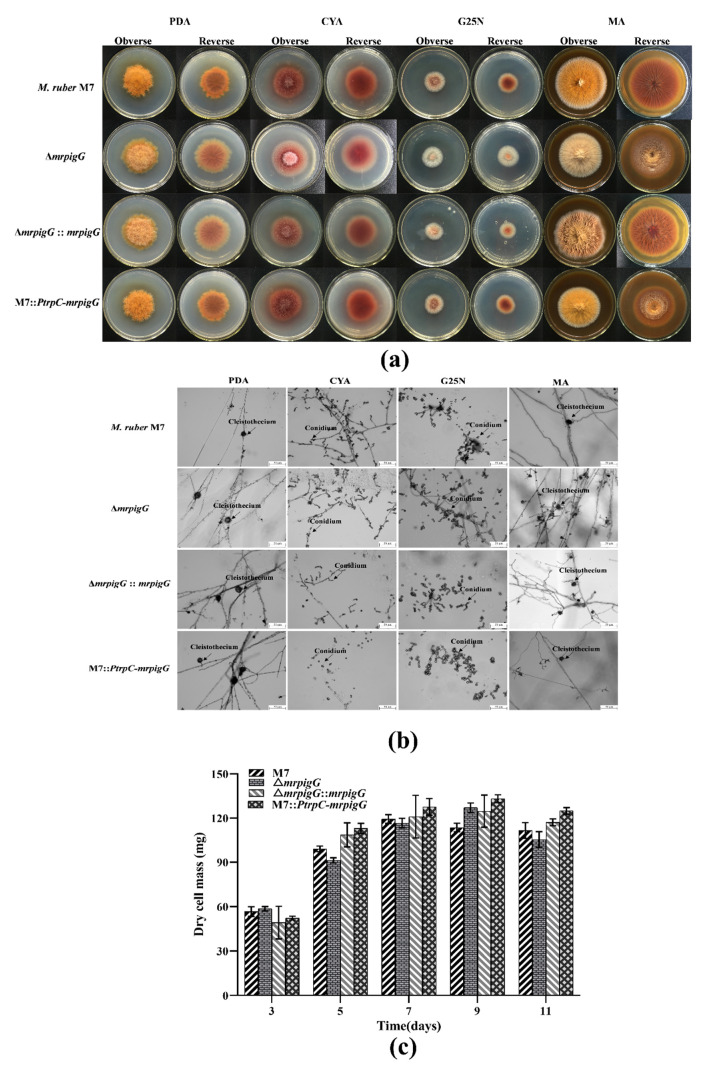
Morphologies and biomasses of Δ*mrpigG*, Δ*mrpigG*::*mrpigG*, M7::*PtrpC*-*mrpigG* and *M. ruber* M7. (**a**) Colonial morphologies on potato dextrose agar (PDA), Czapek yeast extract agar (CYA), malt extract agar (MA) and glycerol nitrate agar (G25N) plates. (**b**) Cleistothecia and conidia formation (at 10 days) of the *M. ruber* M7 and the *mrpigG* transformants on different plates (PDA, CYA, G25N and MA) at 28 °C. (**c**) Biomass (dry cell weight). Experiments were performed in triplicate. Error bars indicate standard deviation.

**Figure 6 jof-06-00156-f006:**
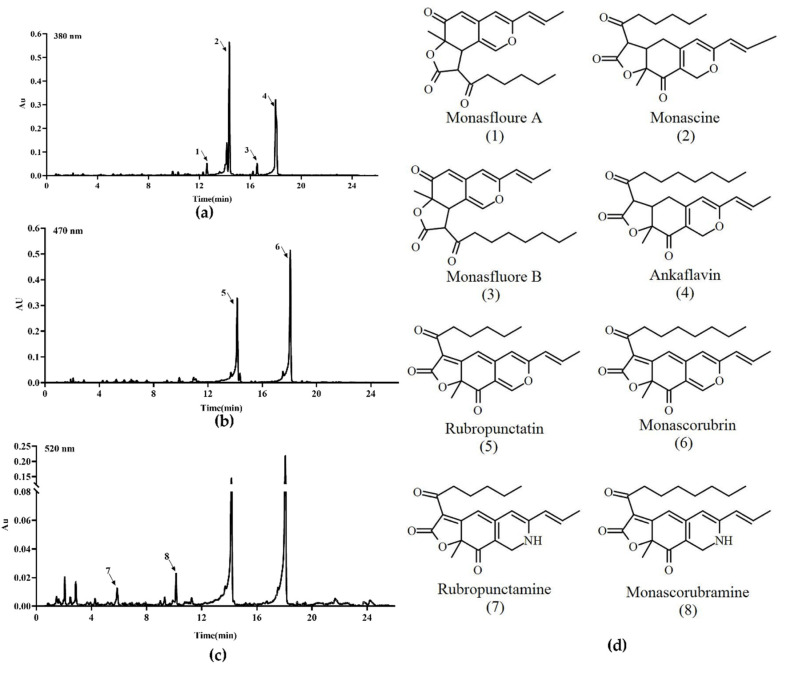
The main pigments of *M. ruber* M7 detected by UPLC. (**a**) The chromatogram of 4 main yellow pigments at 380 nm, which are indicated by 1, 2, 3 and 4. (**b**) The chromatogram of 2 main orange pigments at 470 nm, which are indicated by 5 and 6. (**c**) The chromatogram of the 2 main red pigments at 520 nm, which are indicated by 7 and 8. (**d**) The chemical structure formula of the 8 pigments.

**Figure 7 jof-06-00156-f007:**
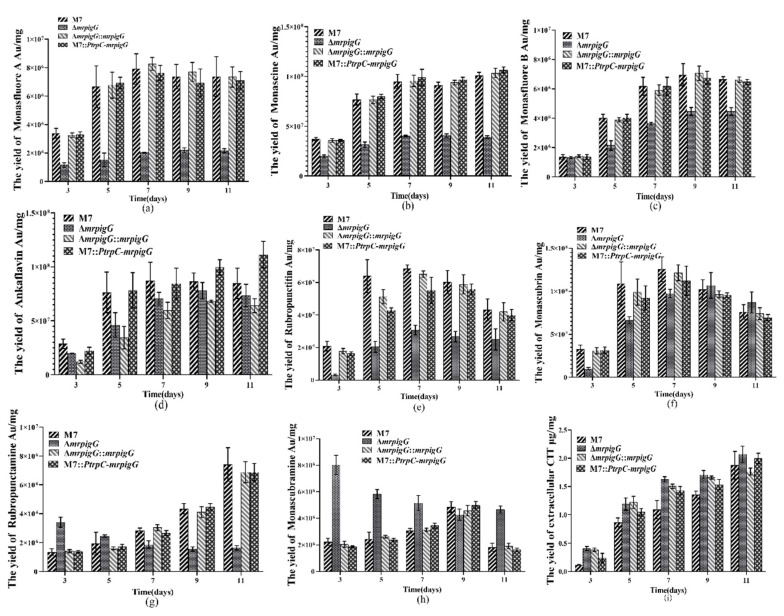
Production of pigments and citrinin by the wild-type strain *M. ruber* M7 and the *mrpigG* transformants. (**a**) The yield of monasfloure A. (**b**) The yield of monascine. (**c**) The yield of monasflore B. (**d**) The yield of ankaflavin. (**e**) The yield of intracellular rubropunctatin. (**f**) The yield of monascorubrin. (**g**) The yield of rubropunctamine. (**h**) The yield of monascorubramine. (**i**) The yield of extracellular citrinin (CIT). The error bar represents the standard deviation between the three repeats. Capitals signify *p*-value < 0.01.

**Figure 8 jof-06-00156-f008:**
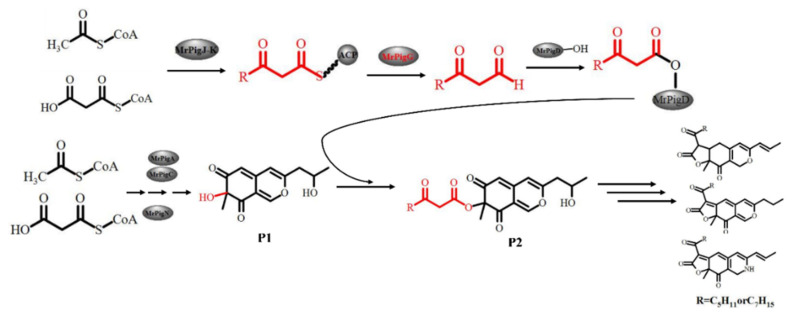
Proposed pathway involving the MrPigG protein.

**Table 1 jof-06-00156-t001:** *Monascus ruber* strains constructed and used in this study.

Strain	Parent	Genotype	Source
M7	M7	Wild-type	Red fermented rice [11]
Δ*mrpigG*	M7	Δ*mrpigG*::*hph*	This study
Δ*mrpigG*:: *mrpigG*	Δ*mrpigG*	Δ*mrpigG*::*mrpigG*-*neo*	This study
M7::*PtrpC*-*mrpigG*	M7	M7::*PtrpC*-*mrpigG*-*neo*	This study

**Table 2 jof-06-00156-t002:** Primers used in this study.

Names	Sequences (5′→3′)	Descriptions
pigG F1	GATCTGCCAGAAATACTAGA	For amplification of the 1236 bp of the whole *mrpigG* gene
pigG R1	TCGGCGAAGCAGAGGCGGAA	
pigG d5F	CGTCCCCCTTCTGCCCAAGA	For amplification of the 884 bp of 5′ flanking regions of the *mrpigG* gene
pigG d5R	CAATATCATCTTCTGTCGAC CCGAACTCCTTGTAGACCGA	
hph F	GTCGACAGAAGATGATATTG	For amplification of the 2137 bp of the *hph* cassette from plasmid pSKH
hph R	CTAGAAAGAAGGATTACCTC	
pigG d3F	GAGGTAATCCTTCTTTCTAGGTGCCGATCAAGACGAAGGA	For amplification of the 867 bp of 3′ flanking regions of the *mrpigG* gene
pigG d3R	CTCTTCCAGCAGGACCAACT	
pigG c5F	CCAGACACCGAACAGCCGCA	For amplification of the 1823 bp of 5′ flanking regions and open reading frame (ORF) of the *mrpigG* gene
pigG c5R	GGTTACGGTTCGATGGGGTTGAGTTGGGTCTCGCTGTAGGGCTGGAT	
G418 F	CCAACTCAACCCCATCGAACCGTAACC	For amplification of the 1221 bp of the *neo* cassette from plasmid pKN1
G418 R	ATCATCATGCAACATGCATG	
pigG c3F	CATGCATGTTGCATGATGATCGATCTTCTTCGCAGACACC	For amplification of the 860 bp of 3′ flanking regions of the *mrpigG* gene
pigG c3R	CGTCACTCGCTTCCAGGTCG	
pigG ov5F	CAGACATACTGCTAAACTCG	For amplification of the 969 bp of 5′ flanking regions of the *mrpigG* gene
pigG ov5R	GGTTACGGTTCGATGGGGTTGAGTTGGGGTGCGGTGCTGGCGAGAGT	
ptrpC F	CATGCATGTTGCATGATGATGTCGACAGAAGATGATATTG	For amplification of the 373 bp of the *ptrpC* promoter from plasmid pKSH
ptrpC R	CATATCGATGCTTGGGTAGA	
*mrpigG* ov3F	TATTCTACCCAAGCATCGATATG ATGCCAGCCAACCGCTCCAG	For amplification of the 1621 bp of 3′ flanking regions and ORF of the *mrpigG* gene
pigG ov3R	TCTTCCAGCAGGACCAACTC	
pigG F2	GCGCTGGCTGCGCTCAT	For amplification of the 503 bp of the partial *mrpigG* gene
pigG R2	CCTCCCACTCCATAACCC	
pigG qF	GGGTGAATGGGCGGGACTA	For real-time qPCR analysis of *mrpigG*(144bp)
pigG qR	GCCAGCAATACGGCAAAGC	
GAPDH F	CAAGCTCACTGGCATGTCTATG	For real-time qPCR analysis of GAPDH (243bp)
GAPDH R	AAGTTCGAGTTGAGGGCGATA	

The sequences underlined by a single straight line are matched with the primer sequences of either hphF or hphR. The sequences underlined by double lines are matched with the primer sequences of G418F or G418R. The sequences underlined by wavy lines are matched with the primer sequences of either PtrpC F or PtrpC R.

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
