# Peer review of "Effects of mrpigG on Development and Secondary Metabolism of Monascus ruber M7"

_jof, 2020, doi:10.3390/jof6030156_

Round 1

Reviewer 1 Report

This was an interesting and beneficial study around the pigments from Monascus species. The study design was appropriate, the methodology well described, and the results interesting. 

Only two minor edits. The first sentence of the introduction is awkward and has some singular/plural issues. A bit or rewording would help.

Table 2 has an arrow in the header that appears out of place

Author Response

Point 1: Only two minor edits. The first sentence of the introduction is awkward and has some singular/plural issues. A bit or rewording would help.

Response 1: the sentence Monascus spp. a kind of medicinal and edible filamentous fungi, have been used ……. for nearly 2,000 years” has been replaced withMonascus spp., as one of medicinal and edible filamentous fungi, have been used ……. for nearly 2,000 years

Point 2: Table 2 has an arrow in the header that appears out of place

Response 2: very thank reviewer for careful reading, I have changed the position of the arrow.

Reviewer 2 Report

This is a well-written paper discussing the role of mrpigG, a secondary metabolism gene involved in the synthesis of Monascus pigments, monacolin K and citrinin (a mycotoxin) in Monascus ruber. To uncover the role of mrpigG in these syntheses, the authors obtain mutant strains of M. ruber (deletion, complemented and overexpressed mrpigG) and recorded the effects on the productions of Monascus pigments and citrinin. Based on the results, I agree with proposed function of mrpigG in biosynthesis pathway. This work thus indeed contribute to furthering the knowledge of this biosynthesis pathway.

My suggestion is to publish the paper, after some minor revisions have been made.

Lines
1 – no space between references (check all the manuscript)
55 – “guessed that MrPigG may contribute to release the very reactive intermediate from” did you mean the intermediate form?
62-67 – I believe this phrase would be much better suited to an abstract rather than to the objective portion of the introduction.
142-147 – I would simplify this paragraph, maybe to something more along the lines of “35:55:10 in the first 3 min to 75:15:10”. Moreover, I believe that there is no need to mention formic acid as solution C. Its commonly used to improve the chromatographic peak in LC analysis, and thus should not affect the overall transition from the polar to apolar phase. If I were righting this portion, I would just give the final concentration of formic acid in the eluent (the 0.1% in water).
148-156 – Same above
187 – substitute the semicolon to full stop in the phrase “ΔmrpigG::mrpigG strains was shown in Figure 2c; A 0.5-kb”
256-257 – I fear this phrase need some restructuring, as it is confusing in its current form.
259- 260 – I would rewrite this phrase as the following “At the end of the 11 day fermentation, extracellular citrinin concentration of the four strains was similar to the wild-type strain M. ruber M7, at 1.88 ± 0.25 µg/mg citrinin”

Author Response

Response to Reviewer 2 Comments

Dear Reviewer

Thanks for your letter and comments on the manuscript entitled “Effects of mrpigG gene on Fungal Development and Secondary Metabolism of Monascus ruber M7” (Manuscript ID: jof-898544). Those comments are all valuable and very helpful for revising and improving this paper. We have studied comments carefully and have made correction which we hope meet with approval.

Point 1:  1 – no space between references (check all the manuscript)

Response 1:  very thank reviewer for careful reading, I have added a space before the references.

Point 2: “guessed that MrPigG may contribute to release the very reactive intermediate from” did you mean the intermediate form?

Response 2: very thank reviewer for careful reading, MrPigG may contribute to release the very reactive intermediate from MrPigA. The reactive intermediate was synthesised by MrPigA (PKS) and attached to its ACP domain.

Point 3: 
 62-67 – I believe this phrase would be much better suited to an abstract rather than to the objective portion of the introduction.

Response 3: thank you for your comment. line 62-67 – The results have revealed that the mrpigG mutants have inconspicuous effects on their morpgologies, growth, MPs and citrinin production except for MPs production in ΔmrpigG, which implies that MrPigG may prefer to biosynthese MP compounds with a five-carbon side chain to ones with a seven-carbon side chain. To our knowledge, it is the first time to report how to control the length of the side chain of MPs by the gene (s) in the MPs gene cluster.

The similar phase has been shown in the abstract, and explained with some detail information for understanding.   Line 17-27 – The results have revealed that the disruption, complementation and overexpression of mrpigG have no apparent defects in morphologies, biomasses and citrinin production except MPs production in ΔmrpigG compared with M. ruber M7. Although the MPs profile of ΔmrpigG and M. ruber M7 was almost same, both mainly including 4 yellow pigments, 2 orange pigments (OP) and 2 red pigments (RP), their yields were decreased in ΔmrpigG in a certain extent. Especially the contents of rubropunctatin (OP)and its derivative rubropunctamine (RP) in ΔmrpigG, both of which have a five-carbon side chain, accounted for 57.7 %, and 22.3 % of those in M. ruber M7. Whereas, monascorubrin (OP) and its derivative monascorubramine (RP) in ΔmrpigG, both of which have a seven-carbon side chain, were increased by 1.15 and 2.55 times. These results suggest that MrPigG may preferentially catalyze the biosynthesis of MPs a five-carbon side chain.

Point 4:  142-147 – I would simplify this paragraph, maybe to something more along the lines of “35:55:10 in the first 3 min to 75:15:10”. Moreover, I believe that there is no need to mention formic acid as solution C. Its commonly used to improve the chromatographic peak in LC analysis, and thus should not affect the overall transition from the polar to apolar phase. If I were righting this portion, I would just give the final concentration of formic acid in the eluent (the 0.1% in water).

Response 4: Thank you for your comment. The original paragraph is “The gradient elution was performed as follows: 35 % (v/v) solvent A , 55 %(v/v) solvent B with10 % (v/v) solvent C maintained for 3 min firstly, the content of solvent A was increased from 35 to 75 % and the solvent B decreased from 55 to 15 %for 15 min, then the solvent A from 75 to 90%, solvent B from 15 to 0 % for 5 min. thirdly, the solvent A decreased from 90 to 35 %, solvent B increased from 0 to 55 % for 5 min. Finally, the column was equilibrated with 35% solvent A,55% solvent B and 10 % solvent C for 3 min.”

The original paragraph has been replaced by “The gradient elution was performed as follows:

firstly, Solvent A/B/C maintained 35:55:10 for 3 min , then the content was changed to 75:15:10 for 15 min, followed by adjusting to 90:0:10 for 5 min, subsequently, the solvent was changed to 35:55:10 for 5 min, finally, the column was equilibrated with 35:55:10 for 3 min.

Point 5: 
 148-156 – Same above (Piont 4)

Response 5: the original phase is “The gradient elution was performed as follows: 90 % (v/v) solvent A with10 % (v/v) solvent B maintained for 3 min firstly. Secondly, the content of solvent A was decreased from 90 to 30 % and the solvent B increased from 10 to 70 %for 7 min. then the solvent A from 30 to 10 %, solvent B from 70 to 90 % for 0.01min and maintained for 3 min. Finally, the solvent A increased from 10 to 90 %, solvent B decreased from 90 to 10 % for 3 min. The temperature of chromatographic column and samples were maintained at 40 ℃ and 4 ℃, respectively”.

The original paragraph has been replaced by “The gradient elution was performed as follows: solvent A/B maintained 90:10 for 3 min firstly. Secondly, the content of solvent A was decreased from 90 to 30 % for 7 min. then the solvent A from 30 to 10 % for 0.01 min and maintained for 3 min. Finally, the solvent A increased from 10 to 90 % for 3 min. The temperature of chromatographic column and samples were maintained at 40 ℃ and 4 ℃, respectively.

Point 6:  187 – substitute the semicolon to full stop in the phrase “ΔmrpigG::mrpigG strains was shown in Figure 2c; A 0.5-kb”

Response 6: very thank reviewer for careful reading. the semicolon was changed to comma —“ΔmrpigG::mrpigG strains was shown in Figure 2c, A 0.5-kb ”

Point 7:  256-257 – I fear this phrase need some restructuring, as it is confusing in its current form.

Response 7: Thank you for your comment. The original sentence is “For extracellular citrinin production, As shown in Figure 7i, the tendency of all strains' citrinin production was always increased from the 3rd day to the 11th day of cultivation in PDA media.”

      The new sentence is “As shown in Figure 7i, the tendency of all strains' extracellular citrinin production was always increased from the 3rd day to the 11th day of cultivation in PDA media.”

Point 8:   259- 260 – I would rewrite this phrase as the following “At the end of the 11 day fermentation, extracellular citrinin concentration of the four strains was similar to the wild-type strain M. ruber M7, at 1.88 ± 0.25 µg/mg citrinin”

Response 8: very thank reviewer for your advice. I have changed the phrase as your recommendation At the end of the 11 days fermentation, extracellular citrinin concentration of the four strains was similar to the wild-type strain M. ruber M7, at 1.88 ± 0.25 µg/mg citrinin”